# Deep Learning Models for Predicting Gas Adsorption Capacity of Nanomaterials

**DOI:** 10.3390/nano12193376

**Published:** 2022-09-27

**Authors:** Wenjing Guo, Jie Liu, Fan Dong, Ru Chen, Jayanti Das, Weigong Ge, Xiaoming Xu, Huixiao Hong

**Affiliations:** 1National Center for Toxicological Research, U.S. Food and Drug Administration, Jefferson, AR 72079, USA; 2Center for Drug Evaluation and Research, U.S. Food and Drug Administration, Silver Spring, MD 20993, USA

**Keywords:** metal–organic framework, gas adsorption, deep learning

## Abstract

Metal–organic frameworks (MOFs), a class of porous nanomaterials, have been widely used in gas adsorption-based applications due to their high porosities and chemical tunability. To facilitate the discovery of high-performance MOFs for different applications, a variety of machine learning models have been developed to predict the gas adsorption capacities of MOFs. Most of the predictive models are developed using traditional machine learning algorithms. However, the continuously increasing sizes of MOF datasets and the complicated relationships between MOFs and their gas adsorption capacities make deep learning a suitable candidate to handle such big data with increased computational power and accuracy. In this study, we developed models for predicting gas adsorption capacities of MOFs using two deep learning algorithms, multilayer perceptron (MLP) and long short-term memory (LSTM) networks, with a hypothetical set of about 130,000 structures of MOFs with methane and carbon dioxide adsorption data at different pressures. The models were evaluated using 10 iterations of 10-fold cross validations and 100 holdout validations. The MLP and LSTM models performed similarly with high prediction accuracy. The models for predicting gas adsorption at a higher pressure outperformed the models for predicting gas adsorption at a lower pressure. The deep learning models are more accurate than the random forest models reported in the literature, especially for predicting gas adsorption capacities at low pressures. Our results demonstrated that deep learning algorithms have a great potential to generate models that can accurately predict the gas adsorption capacities of MOFs.

## 1. Introduction

Nanomaterials have been widely studied in various fields, such as food science, energy, electronics, and drugs, due to their advantageous physical, chemical, optical, and electrical properties [1,2,3,4,5,6]. In contrast to other rigid nanoparticle carriers, metal–organic frameworks (MOFs) have gained attention in recent years due to their well-defined structure, ultrahigh surface area, high porosity, tunable pore size, and easy chemical functionalization. In a MOF, metal ions or clusters of ions are linked by organic molecules to form a repeating, cage-like hollow structure. The unique structure diversity and exceptionally large surface area make MOFs promising candidates for many applications including intracellular delivery of drugs, proteins, and nucleic acids; sensing; bioimaging; and energy storage [7]. In contrast to other rigid nanoparticles, MOF nanocarriers have a high drug loading capacity and controlled drug release properties, which contributes to the growing attention on MOFs for pharmaceutical applications [8]. 

The industrial and biomedical applications using MOFs have increased over the past few years, including catalysis, gas storage, and gas separation. By carefully selecting metal clusters and organic linkers, researchers can synthesize MOFs to selectively adsorb specific gases [9,10,11]. To find good performing materials for some specific applications, gas adsorption data are typically used to screen a diverse MOF database. Due to the large number of possible MOFs, experimentally it is not practical to generate all possible MOFs and measure their adsorption capacities for a variety of gases. Therefore, computational methods, such as grand canonical Monte Carlo (GCMC) simulation and molecular dynamics simulation, have been used as alternative methods [12,13,14,15]. However, even with high performance computing technologies, current computational methods, such as GCMC simulations, are very computationally intensive. Sometimes GCMC simulations can be practically infeasible, depending on parameters, such as the size of the MOF database, the number of gases, the operating conditions (temperatures/pressure conditions), and the number of compositions. Therefore, other fast and reliable computational methods are urgently needed to predict gas adsorption capacities of MOFs.

A variety of computational methods have been developed and used for predicting biological activities and physicochemical properties of chemicals [16,17,18,19,20,21,22,23]. For example, our group have used machine learning as an attractive computational technique to provide alternative methods for estimating physicochemical properties and toxicological activities of chemicals [24,25,26,27,28,29,30,31]. Machine learning methods are also effective tools that can reveal the underlying structure–property relationship, and hence, accurately predict gas adsorption capacities of MOFs [12,15,32,33]. For machine learning applications, a large amount of accurate data from either experiments or computational methods is needed to build reliable models. Since experimental data are generally restricted to a small number of MOFs under limited experimental conditions, computational prediction seems a better source for large quantities of data. Thus, combinations of GCMC simulations and machine learning have been widely used to improve the efficiency of predicting gas adsorption over various conditions [34,35,36]. The combination allows GCMC to focus on the most promising materials, and machine learning could be trained to predict gas adsorption of MOFs, which are not covered by the GCMC simulations. Machine learning can minimize the computationally intensive GCMC simulations by predicting the gas adsorption capacities from the topology of MOFs, thereby replacing the GCMC simulations. With large numbers of structures and limited gas adsorption data, there is a high likelihood that the incorporation of machine learning methods could help design and develop MOFs with the desired gas adsorption capacities.

The continuously increasing sizes of MOF datasets and the complicated relationships between MOFs and their gas adsorption capacities make deep learning a suitable approach. Some deep learning models have been developed to predict the gas adsorption capacities of MOFs. For example, Lee et al. developed models using deep learning and evolutionary algorithms to find MOFs with desired properties from an extremely diverse and large MOF dataset containing over 100 trillion structures [37]. The deep learning models were shown to be able to discover the high-performing MOFs with optimal working capacity [37]. Anderson et al. demonstrated that by leveraging a large amount of GCMC simulation data, deep neural network models were trained to predict hydrogen volumetric adsorptions at various temperatures and pressures [13]. In their later work, Anderson et al. trained deep learning models to predict the full adsorption isotherm for molecules such as methane, nitrogen, xenon, and krypton, using both geometric and chemical descriptors [38]. Their results showed that deep learning models can be used to predict the adsorption of different adsorbates at different operating conditions for new MOFs which are not included in the training set. In another example, Ma et al. trained a deep neural network model with two hidden layers on 13,506 MOFs to predict H_2_ adsorption at 100 bar and 243 K [39]. The coefficient of determination (R^2^) obtained was 0.998, suggesting deep learning is a very promising technique to study H_2_ adsorption properties of MOFs. Wang et al. developed a graph convolutional neural network to screen high-performing MOFs from a large database based on the data generated by GCMC simulation [40]. The area under the receiver operating characteristic curve of 0.93 demonstrated the reliability and accuracy of the developed deep learning models. In addition, the incorporation of deep learning models reduced the time used to evaluate several hundreds of thousands of hypothetical MOFs (hMOFs).

In this study, we used two deep learning algorithms, multilayer perceptron (MLP) and long short-term memory (LSTM) networks, to predict the gas adsorption capacities in MOFs. The MOF datasets were obtained from the Northwestern hMOFs database that was computationally designed by Wilmer et al. [41]. The MOF structures were represented by structural descriptors (dominant pore size, max pore size, void fraction, gravimetric surface area, and density) [41] and chemical descriptors (atom number density) [36]. First, the deep learning models were trained on the datasets of methane and carbon dioxide adsorption at various pressures. The models were evaluated using 10 iterations of 10-fold cross validations and 100 holdout validations. We used the same validation methods as previous studies [36] to directly compare our deep learning models with previously reported machine learning models. The random forest models developed by Fanourgakis et al. [36] have greatly improved the prediction accuracy of gas adsorptions of MOFs. Their models significantly outperformed the previously reported models [35,42] where only structural descriptors were used. By directly comparing our results with the results from the random forest models, we demonstrated the potential of our deep learning models to accurately predict the gas adsorption capacities of MOFs. 

## 2. Materials and Methods

### 2.1. Data Preparation

The workflow for developing deep learning models is shown in Appendix A. In this study, the hMOFs database developed by Wilmer and coworkers [41] was used to build deep learning models (MLP and LSTM) for predicting gas adsorption capacities of MOFs. 

In the database, 102 building blocks and 15 functional groups were geometrically assembled using a bottom-up construction algorithm such as snapping Tinkertoy. There are 137,953 hMOFs generated using this Tinkertoy algorithm. The detailed structures of hMOFs could be found at https://mof.tech.northwestern.edu/ (accessed on 8 April 2021) along with the carbon dioxide and methane adsorption capacities at *T* = 298 K and at several pressures. The gas adsorption capacities of methane at 1, 5.8, 35, and 65 bar, and of carbon dioxide at 0.05, 0.5, and 2.5 bar, for these MOFs were calculated using GCMC simulations [43]. In addition to MOFs, a covalent organic frameworks (COFs) dataset [44] and their methane adsorption capacities at 65 bar were also used as an independent dataset to validate model performance.

To describe the structures of MOFs, five geometric descriptors (the dominant pore size, the max pore size, the void fraction, the gravimetric surface area, and density) taken from the Northwestern University database were used in this study. The detailed calculation for structural descriptors can be found in the work of Wilmer et al. [45]. Another structural descriptor of MOFs, accessible volume, was also calculated using Zeo++ code [46] with a probe radius of 1.625 Å and 50,000 Monte Carlo samples per unit cell. Twenty atom types (atoms, namely, H, C, N, O, F, Cl, Cu, Zn, Br, and Zr, and their hybridization and connectivity types) were used to characterize the chemical environment of the pores in MOFs. For example, C_1, C_2, C_3, and C_R were used to represent carbon with single, double, triple, and aromatic bonds, respectively. A similar naming convention was used for nitrogen (N_1, N_2, N_3, N_R) and oxygen (O_1, O_2, O_3, O_R). The atom number density was defined as the number of atoms for each atom type in a unit cell of MOFs. The atom number density was calculated by Fanourgakis et al. using the Python program lammps_interface [36]. MOFs with unidentified atom types were removed from the datasets. Details for the final eight datasets used in this study are provided in Table 1. 

### 2.2. Deep Learning Algorithms

MLP is a feedforward neural network that utilizes a supervised learning technique called backpropagation to recognize underlying relationships in data. The basic structure of an MLP consists of an input layer, one or more hidden layers, and an output layer. Each layer is made up of neurons and each neuron is connected to all the neurons in the next layer by weight. The weights are randomly chosen at the beginning of a training process and then are calculated from the backpropagation process to minimize errors between predicted values from the output layers and the actual values. The input values are transformed to the output signals by an activation function. In this study, we used the rectified linear unit (ReLU) activation function because of its capability to quickly converge. After the MLP model was trained with established network topology and the final sets of weights, the model was ready to make predictions.

To demonstrate the power of deep learning, the LSTM algorithm was also used in this study to build models for predicting the gas adsorption capacities of MOFs. LSTM is an advanced recurrent neural network and has been widely used for natural language processing due to its ability to deal with sequential data [47,48]. LSTM uses a special unit called a memory cell that controls the memorizing process. Each LSTM memory cell contains three gates: an input gate, an output gate, and a forget gate. The input gate passes the new information from the input to the cell, and the forget gate controls whether the information from the previous timestamp is needed and should be passed to the cell or is irrelevant and should be ignored. Finally, the updated information from the cell is passed to the next timestamp through the output gate. Besides sequential data, LSTM also proves to be powerful in processing nonsequential data. In this work, although the structural and chemical descriptors are not in the form of sequences, they were processed as fixed-length vectors and used to train the LSTM models. 

### 2.3. Model Development

Since hyperparameters are tunable and have a direct impact on the model performance, hyperparameter optimization is the first step in model development to search for hyperparameters that could maximize the predictive accuracy of the model. To prevent information leaking in the training models, the training dataset was randomly split into five subsets. This random splitting was repeated five times. For each splitting, four subsets were used to build predictive models with different sets (combinations) of hyperparameters. The gas adsorption capacities of MOFs in the remaining subset were predicted using the built models. The process was repeated so that each of the five subsets was used only once as a testing dataset. The testing results yielded from the models built with the same set of hyperparameters were averaged to measure the performance of the models constructed using the set of hyperparameters. The set of hyperparameters that had the highest average prediction accuracy in the testing was selected to construct a model on the training dataset. 

In training LSTM models, four hyperparameters (number of neurons, number of layers, batch size, and number of epochs) were optimized using the abovementioned procedures. In training MLP models, three hyperparameters (number of layers, number of neurons, and alpha values for regularization) were tuned using the same procedure. Appendix A provide examples of tuning hyperparameters in LSTM and MLP models, respectively, using a dataset containing 5000 MOFs and 5000 COFs. 

### 2.4. Model Evaluation

To evaluate the performance of MLP and LSTM deep learning models, 10 iterations of 10-fold cross validations and 100 holdout validations (also called external validations in some literatures) were conducted.

As shown in Appendix A, in a 10-fold cross validation, a dataset was randomly divided into 10 groups using a fixed random seed. Nine groups were used to build MLP and LSTM models and the remaining group was then used to evaluate the constructed models. This process was iterated 10 times so that each of the 10 groups was used only once as the test set. The predicted adsorption capacities of MOFs for gas from the 10 models for an algorithm (MLP or LSTM) were compared with the values from GCMC simulations to calculate the performance metrics for the evaluation of the models. To reach a statistically robust estimation of model performance, the 10-fold cross validation was repeated 10 times by randomly dividing the whole dataset into 10 groups using 10 different random seeds. 

As shown in Appendix A, in a holdout validation, a dataset was randomly split into two sets: a training set with 10,000 MOFs and a test set containing the rest of the MOFs. The training set was used to train MLP and LSTM models, and the test set was used to evaluate the performance of the models. The holdout validation was repeated 100 times by randomly splitting a dataset into training sets and test sets using 100 different random seeds. 

The performance of the MLP and LSTM models was measured by Pearson correlation coefficient (*r*), *R*^2^, root mean square error (RMSE), and mean absolute error (MAE). The performance metrics *r*^2^, *R*^2^, RMSE, and MAE were computed using Equations (1)–(4).
(1)$$r=\frac{n\sum_{1}^{n}y_{i}u_{i}-\sum_{1}^{n}y_{i}\sum_{1}^{n}u_{i}}{\sqrt{\left(n\sum_{1}^{n}y_{i}{}^{2}-{\left(\sum_{i}^{n}y_{i}\right)}^{2}\right)(n\sum_{1}^{n}u_{i}{}^{2}-{\left(\sum_{i}^{n}u_{i}\right)}^{2})}}$$
(2)$$R^{2}=1-\frac{\sum_{1}^{n}{\left(y_{i}-u_{i}\right)}^{2}}{\sum_{1}^{n}{\left(y_{i}-z\right)}^{2}}$$
(3)$$RMSE=\sqrt{\frac{\sum_{1}^{n}{\left(y_{i}-u_{i}\right)}^{2}}{n}}$$
(4)$$MAE=\frac{\sum_{i}^{n}\left|y_{i}-u_{i}\right|}{n}$$
where, *y_i_* is the observed gas adsorption value for the MOF *i*; *z* is the average observed gas adsorption value of MOFs predicted; *u_i_* is the predicted gas adsorption for the MOF *i*; *n* is the number of MOFs predicted.

Since *RMSE* and MAE are scale-dependent metrics, they are not suitable for comparing the performance of models for predicting dependent variables with values of different scales such as gas adsorption values at different pressures. Thus, scale independent metrics, scaled RMSE (sRMSE) and scaled MAE (sMAE), were defined in Equations (5) and (6) and used in this study.
(5)$$sRMSE=\frac{RMSE}{z}$$
(6)$$sMAE=\frac{MAE}{z}$$

## 3. Results and Discussion

### 3.1. Prediction of Methane Adsorption of MOFs

The deep learning algorithms MLP and LSTM were used to build models for predicting adsorption capacities of methane at pressures of 1, 5.8, 35, and 65 bar. The performance of the models was evaluated using 10 iterations of 10-fold cross validations and 100 repeats of holdout validations using *r*^2^, *R*^2^, sRMSE, and sMAE.

The predicted gas adsorptions of MOFs from LSTM and MLP models in the 10-fold cross validations were compared with their actual gas adsorptions in Appendix A. Figure 1 summarizes the performance of LSTM and MLP models on four datasets of methane adsorption capacities at pressure 1, 5.8, 35, and 65 bar from the 10 iterations of 10-fold cross validations. For comparison, the *r*^2^, sMAE, and sRMSE of the 10-fold cross validations of random forest models on the same datasets [36] are included as blue bars in Figure 1B–D, respectively. The models generated from the dataset of methane adsorption at a higher pressure outperformed the models yielded from the dataset of methane adsorption at a lower pressure, regardless of which performance metric (*r*^2^, sMAE, and sRMSE) or deep learning algorithm (MLP and LSTM) were used. More specifically, the average *R*^2^, *r*^2^, sMAE, and sRMSE values for the MLP models built from the dataset at 65 bar are 0.9652, 0.9661, 0.0379, and 0.05380, respectively. The corresponding metrics values for the MLP models constructed using the dataset at 1 bar are 0.9066, 0.9126, 0.1292, and 0.2945, respectively. A similar trend was observed for the LSTM models and the previously reported random forest models. At high pressures, the methane adsorption capacity of MOFs was found to correlate strongly with geometrical descriptors, such as pore sizes, void fractions, and surface area [36,45,49,50]. The adsorption capacity of activated carbon dioxide was positively correlated with surface area and pore volume. At high pressures, the enlarged surface area and high porosity provide more sites for gas adsorption [51]. Since these geometric descriptors (the dominant pore size, the max pore size, the void fraction, the gravimetric surface area, and density) were used in our study to characterize MOFs, our model accurately predicted the methane adsorption capacities. Methane adsorption at 5.8 and 65 bar is useful for on-board vehicular natural gas storage technologies [52], and our deep learning models can be used to help identify high methane adsorption MOFs for such applications.

For predictions at 1 bar, *r*^2^ for MLP and LSTM are 0.9126 and 0.9126, respectively, which are substantially better than the 0.886 reported for random forest models [36]. At low pressures, the strength of interactions between adsorbent molecules and MOFs plays a more important role in the adsorption capacity of MOFs. We speculate that the adsorption capacities of MOFs at high pressures are mainly determined by the shapes of structures of the MOFs as well as the accessible pore volumes and internal surface areas of MOFs, which are easy to characterize for MOFs. Therefore, adsorption capacities at high pressures can be accurately predicted using both traditional machine learning and deep learning algorithms. However, the chemical properties, such as hydrophobic and electrostatic interactions between gases and MOFs at the atomic level, contribute more to the adsorption capacities of MOFs for gases at low pressures. The interactions at the atomic level between gases and MOFs are more difficult to model. Thus, the methods that can recognize complicated relationships, such as deep learning algorithms, could better model gas adsorption of MOFs at low pressures. The improved prediction accuracy of our deep learning models demonstrates that deep learning is a more suitable approach than traditional machine learning algorithms to address complex relationships between methane adsorption and MOF structures at low pressures. The performances of the two deep learning models were similar for predicting methane adsorptions at all pressures. 

Another challenge in predicting the adsorption capacities of MOFs using machine learning and deep learning is the lack of experimental data for training. Experimentally determined adsorption capacities are the best for training machine learning and deep learning models. However, to experimentally generate adsorption data for such a large number of MOFs is time consuming and costly, making it very practically challenging, if not impossible. Therefore, most of the current practices use the adsorption capacities calculated by GCMC, which is a force-field method based on Henry’s law. There has historically been a major challenge of developing accurate force-fields for describing adsorption, especially for adsorption at low pressures based on Henry’s law. In brief, Henry’s law states that the adsorption capacity of a MOF is proportional to the gas pressure at a constant, which can be determined experimentally or with a force-field-based method. Usually, at high pressures, the effect of gas pressure overpasses the contributions from chemical and energetic interactions between the MOF and the gas. Therefore, the adsorption capacities at high pressures estimated from a force-field-based method are close to that determined by experiments. However, at low pressures, chemical and energetic interactions between the MOF and the gas are important to the adsorption capacity and are not linearly proportional to the gas pressure. Thus, the adsorption capacities at low pressures estimated using a general force-field method may have a large difference from that determined experimentally. Though the deep learning models improved the prediction performance on adsorption capacities at low pressures, we should be cautious in the utilization of the results because the predicted values are fitted to the adsorption capacities estimated by GCMC but not to that determined experimentally.

Since a representative low pressure for natural gas delivery in a vehicle engine is 5 bar, MOFs that have high methane adsorption capacities at a pressure of 1 to 5 bar are strongly needed to facilitate the transport industry [53]. Our deep learning models can help identify high-performing MOFs for methane storage at low pressures. Compared to the models for predicting methane adsorption at high pressures, the models for predicting methane adsorption at low pressures need to be improved in the future. Energy descriptors have been used in recent studies [34,35,54] to improve predictions in the low-pressure regime. For example, Fanourgakis et al. [35] used potential energy surface as descriptors and evaluated the model performance on 4764 computation-ready, experimental (CoRE) MOFs. Bucior et al. [54] used sorbate–sorbent energy histograms as descriptors and tested them on more than 50,000 experimental MOFs. Deng et al. [34] used the heat of adsorption as the energy descriptors to screen 6013 CoRE-MOFs. Therefore, the deep learning models for predicting the gas adsorption of MOFs at low pressures are expected to be improved by including chemical and energetic descriptors. The accurate deep learning models for methane adsorption at low pressures are also helpful to explore the adsorption and delivery of biological gases, such as nitric oxide (NO), carbon monoxide (CO), and hydrogen sulfide (H_2_S), for biomedical applications [55]. 

In the holdout validations, 10,000 MOFs were used to train deep learning models and the remaining MOFs (60,608, 18,417, 60,605, and 15,151 for adsorption at pressure 1, 5.8, 35, and 65 bar, respectively) were used to evaluate the trained models. The prediction results of the MLP and LSTM models on the four datasets are provided in Appendix A. Figure 2 shows the distributions of performance metrics *r*^2^, *R*^2^, sRMSE, and sMAE of MLP and LSTM models. As shown in the figure, the MLP and LSTM models performed similarly. The average *R*^2^ values for predicting methane adsorptions at 1, 5.8, 35, and 65 bar are 0.8722, 0.9441, 0.9556, and 0.9602, respectively, for the LSTM models, and 0.8755, 0.9475, 0.9544, and 0.9592, respectively, for the MLP models. Moreover, the models for predicting adsorption at a higher pressure had smaller ranges of performance metrics than the models for predicting adsorption at a lower pressure, consistent with the observations from 10-fold cross validations. The comparison between our deep learning models and the random forest models (dashed vertical lines) reported in the literature [36] shows that the deep learning models had higher *r*^2^ as well as lower sMAE and sRMSE values (Figure 2B–D), especially for models for predicting methane adsorption at low pressures.

To examine if the deep learning models could be improved when including more rudimentary information, such as accessible pore volumes, we calculated the accessible pore volumes using Zeo++ (http://zeoplusplus.org/, accessed on 9 September 2022) with the probe radius of 1.625 Å. We then developed MLP and LSTM models for predicting methane adsorption at 1 bar by integrating the accessible pore volumes into the datasets and using the same learning protocol. We also developed MLP and LSTM models using datasets with the removal of atom types and the inclusion of accessible pore volumes. The results from 100 holdout validations on the models with all 26 descriptors were summarized in Appendix A, and the models developed with different sets of descriptors were compared in the Appendix A. As can be seen from the comparison, the descriptors of atom types are important for predicting methane adsorptions at low pressures because the models without atom types greatly underperformed, while the inclusion of accessible volumes did not improve the deep learning models. Because accessible volumes are correlated with surface areas, our results indicate that adding correlated descriptors would not much improve deep learning models for predicting the gas adsorption of MOFs.

When examining the MOFs that showed large differences between the deep learning models and GCMC-predicted adsorptions at modest pressures, we found that MOFs with smaller accessible pores, less surface area, and greater weight density had larger disagreements between the deep learning model-predicted and GCMC-calculated adsorption capacities. For example, comparing the deep learning-predicted methane adsorption capacities at 5.8 bar with the GCMC-calculated values (Appendix A) revealed that the 1775 MOFs with over 100% overpredictions have an average dominant pore diameter of 4.35 Å, gravimetric surface area of 969.57 m^2^/g, and weight density of 1.37 g/cm^3^, while the 4542 MOFs with less than 5% disagreements have an average dominant pore diameter of 8.76 Å, gravimetric surface area of 3267.07 m^2^/g, and weight density of 0.74 g/cm^3^. As gas adsorption is mostly driven by dispersion at modest pressures, the MOFs with great density usually have small pores and surface areas, which makes gas dispersion more difficult, and adsorption capacities are determined by multiple chemical and structural features in a complicated relationship. Therefore, predicting the adsorption capacities of such MOFs at modest pressures is more challenging than of MOFs with low density, large pores, and large surface areas. Though we demonstrated that the developed deep learning models are helpful in screening a large dataset, the utilization of the deep learning prediction results for MOFs with great density, small pores, and surface areas should be cautious.

### 3.2. Prediction of Carbon Dioxide Adsorption of MOFs

Deep learning models were developed using MLP and LSTM algorithms for predicting carbon dioxide adsorption at pressures of 0.05, 0.5, and 2.5 bar. Model performance was evaluated using 10 iterations of 10-fold cross validations and 100 repetitions of holdout validations and measured by *r*^2^, *R*^2^, sRMSE, and sMAE.

Appendix A show the prediction results from the 10 iterations of 10-fold cross validation for LSTM and MLP models based on the three datasets of carbon dioxide adsorption capacities of MOFs at pressures of 0.05, 0.5, and 2.5 bar. The performance metrics calculated from the prediction results are summarized in Figure 3. The MLP and LSTM models performed similarly with high prediction accuracy. For predicting carbon dioxide adsorption at 2.5 bar, the average *R*^2^, *r*^2^, sMAE, and sRMSE values are 0.9621, 0.9635, 0.0803, and 0.1164, respectively, for the MLP models, and 0.9572, 0.9591, 0.0839, and 0.1231, respectively, for the LSTM models. The results indicate the used chemical and structural descriptors are related to carbon dioxide adsorption capacities at high pressures. At a high pressure, carbon dioxide adsorption is driven by physical interactions instead of chemical interactions [56]. Therefore, geometric descriptors used in this study might be the major contributors to the good performance of the developed deep learning models. Highly accurate predictions of carbon dioxide adsorption at high pressures could facilitate the design of appropriate MOFs for the precombustion of carbon capture to efficiently reduce carbon dioxide emission from fossil fuels before the combustion is completed. In addition to environmental protection, the accurate deep learning models for carbon dioxide adsorption at relatively high pressures could also be extended to facilitate the design of MOFs to adsorb other gases, such as Xe and Kr, which are in high demand in the medical industry. 

When predicting carbon dioxide adsorption at 0.05 bar, the MLP and LSTM models achieved *r*^2^ values of 0.7816 and 0.8022, respectively, which are substantially higher than 0.752 from the random forest models reported in the literature [36]. The most studied carbon capture and storage is the post-combustion approach, where carbon dioxide is adsorbed from flue gas at a low pressure (including atmospheric pressure). Thus, it is particularly promising to have accurate models for the identification of MOFs with high adsorption capacities at low pressures. Meanwhile, compared to the models at high pressures, the deep learning models can be significantly improved by including well-characterized descriptors. In comparison to methane, the interactions between carbon dioxide and MOFs are more complex due to the quadrupolar interactions in carbon dioxide molecules. Therefore, it is more demanding for carbon dioxide adsorptions to use descriptors that are more related to carbon dioxide adsorption.

The predictions from the 100 holdout validations were compared to the actual adsorptions of the MOFs in Appendix A. Figure 4 summarizes the performance of the MLP and LSTM models in the 100 holdout validations. Similar to the methane adsorption models, the MLP and LSTM models performed equally well in the prediction of carbon dioxide adsorption of MOFs, especially for predicting adsorption at high pressures, in terms of all four metrics. For example, for predicting adsorption at 2.5 bar, the LSTM and MLP models had equally high average *R*^2^ values of 0.9434 and 0.9494, respectively. Furthermore, a similar trend was observed: the models for predicting carbon dioxide adsorption at a higher pressure outperformed the models for predicting adsorption at a lower pressure. When compared to the reported random forest models [36] (dashed vertical lines in Figure 4B–D), the MLP and LSTM models had higher *r*^2^ and lower sMAE and sRMSE values, indicating that the deep learning models outperformed the traditional machine learning random forest models, especially for predicting carbon dioxide adsorption at low pressures. Although the MLP and LSTM models performed better than the random forest models [36] in the prediction of carbon dioxide adsorption at 0.05 bar, their performances were much poorer than the corresponding MLP and LSTM models for predicting adsorptions at higher pressures, with much lower average *r*^2^ and *R*^2^ values and higher sMAE and sRMSE. Moreover, variation in model performances for predicting carbon dioxide adsorption at a low pressure was larger than the corresponding one at a high pressure (a wider distribution of performance metric values), consistent with the observation in the 10-fold cross-validations. Therefore, more informative descriptors are needed to improve deep learning for the prediction of carbon dioxide adsorption at low pressures.

### 3.3. Model Transferability

The 27,151 MOFs and 17,098 COFs [44], as well as their methane adsorption data at 65 bar, were used to evaluate the transferability of MLP and LSTM models. MLP and LSTM models were built on one type of nanomaterials (MOFs or COFs) and then evaluated on the other type of nanomaterials (COFs or MOFs). The results are given in Figure 5.

As shown in Figure 5, the MLP and LSTM models trained on the MOFs accurately predicted methane adsorptions of the COFs (the red points), with high *r*^2^ values of 0.9270 and 0.9514, respectively, indicating that the deep learning models generated from MOFs can be used to predict the methane adsorption of COFs. In contrast, the MLP and LSTM models trained on the COFs had much worse predictions on the MOFs (the blue points), resulting in much lower *r*^2^ values of 0.5281 and 0.4365, respectively. Furthermore, the predicted adsorption values are larger than the actual adsorption data for most of the 27,151 MOFs. 

The observed higher transferability for the MLP and LSTM models trained on MOFs than for the models trained on COFs could be associated with the larger size and wider range of adsorption data; thus, more diverse structures of MOFs than COFs (Table 1), at least partially. 

The significantly larger predicted adsorptions compared to the actual data (the blue points in Figure 5) could be caused by the structural differences between MOFs and COFs. MOFs contain metal clusters, while COFs are made entirely from light elements, such as H, B, C, N, and O. Therefore, the MLP and LSTM models trained on COFs do not have information on heavy atoms, which are related to methane adsorption of MOFs, leading to less accurate predictions from such models. In contrast, the MLP and LSTM models trained on MOFs convey information on both heavy and light atoms, and thus, can be well transferred to adsorption predictions of COFs. Among the 25 descriptors, 19 were found to have smaller values for the COFs than that for the MOFs. For example, the average mass density is 0.24 g/cm^3^ for the COFs, smaller than the 0.86 g/cm^3^ for MOFs. The atom density for oxygen with double bonds is 0.09 for COFs, which is again smaller than the 5.71 for MOFs. Since MOFs and COFs have similar adsorption capacities, the model built on COFs data with relatively smaller descriptors values would overestimate the adsorption in MOFs.

### 3.4. Models Constructed Using a Mixture of MOFs and COFs

The above-described deep learning models were developed on the same type of nanomaterials, MOFs or COFs. To examine the capability of deep learning to develop accurate gas adsorption prediction models using multiple types of nanomaterials, we developed and evaluated MLP and LSTM models using a mixture of 17,098 COFs and 27,151 MOFs. The model performance was evaluated using 100 holdout validations. In a holdout validation, 5000 COFs and 5000 MOFs were randomly selected as the training dataset to build MLP and LSTM models, and the remaining COFs and MOFs were used as the test set to evaluate the performance of the MLP and LSTM models. This procedure was repeated 100 times and the resulting 100 model performance measurements were used to make a statistically robust evaluation of the MLP and LSTM models. 

The methane adsorption predictions for the COFs and MOFs were compared in Appendix A. The distributions of the performance metrics (*R*^2^, *r*^2^, sMAE, and sRMSE) from the 100 holdout validations are shown in Figure 6. The LSTM model and MLP models had similarly high prediction accuracies, average *r*^2^ = 0.9678, *R*^2^ = 0.9660, sMAE = 0.0329, and sRMSE = 0.0514 for the LSTM models, and average *r*^2^ = 0.9660, *R*^2^ = 0.9638, sMAE = 0.0350, and sRMSE = 0.0532 for the MLP models. Moreover, these MLP and LSTM models performed better than the reported random forest model, *r*^2^ = 0.965, demonstrating that deep learning can generate more accurate models based on mixtures of nanomaterials for predicting methane adsorption capacities. 

The prediction accuracies of the MLP and LSTM models developed using mixtures are close to those of the models developed using only MOFs (the blue curves in Figure 2). For example, for predicting methane adsorption at 65 bar, the 100 holdout validations for the LSTM and MLP models developed on MOFs yielded average *r*^2^ values of 0.9628, and 0.9624, respectively. The LSTM and MLP models built on COFs also had similarly high performances in the 100 holdout validations with average *r*^2^ values of 0.9862 and 0.9866, respectively. Although the structures of MOFs and COFs are different, 5000 structures of MOFs or COFs are large enough to cover the similar structural spaces of 10,000 MOFs or COFs. Thus, the models trained with a mixture of 5000 MOFs and 5000 COFs performed equally as well as the models trained with 10,000 MOFs or COFs. Our results suggest that increasing the number of MOFs or COFs to greater than 5000 would not much improve the trained deep learning models.

## 4. Conclusions

In the previous work of Fanourgakis et al. [36], they introduced a new chemical descriptor “atom type” in the machine learning algorithms to account for the chemical characters of MOFs. They evaluated the traditional machine learning models (random forest) on different gases (carbon dioxide, methane) and found that using the new descriptors could significantly improve the prediction accuracy. In our work, we took a step further to investigate whether deep learning models could further improve the prediction of gas adsorption capacities. We developed MLP and LSTM models to predict gas adsorption capacities in a large number of topologically diverse MOFs and COFs. Both structural descriptors and chemical descriptors were used to characterize the MOFs and COFs. The performance of the deep learning models built with MLP and LSTM were evaluated using 10-fold cross validations and holdout validations. Our results confirmed that models with both structural and chemical descriptors could accurately predict the gas adsorption capacities of MOFs. We also found that deep learning models can further improve the prediction accuracy by handling the complex relationship between gas adsorption capacities and MOF structures, especially at low pressures. The LSTM and MLP models performed equally well for predicting methane and carbon dioxide adsorptions at various pressures. The deep learning models for predicting adsorption of gases at high pressures performed better than the models for predicting adsorption at low pressures. However, when comparing to the performance at high pressures, the performance at low pressures was still relatively low, indicating that the use of deep learning models are not sufficient to improve predictions. Therefore, independent descriptors more related to gas adsorption are needed to improve deep learning models for predicting gas adsorption at low pressures. The comparison of the MLP and LSTM models with the reported random forest models [36] demonstrated that the deep learning models can accurately predict gas adsorption of MOFs and COFs and assist in the designing of highly performing MOFs and COFs for gas adsorption. Furthermore, we demonstrated the deep learning models built on MOFs can be used to predict gas adsorption of other nanomaterials, such as COFs. Our results also indicate that accurate deep learning models can be developed using mixtures, demonstrating the feasibility of integrating diverse nanomaterials for deep learning.

## Figures and Tables

**Figure 1 nanomaterials-12-03376-f001:**
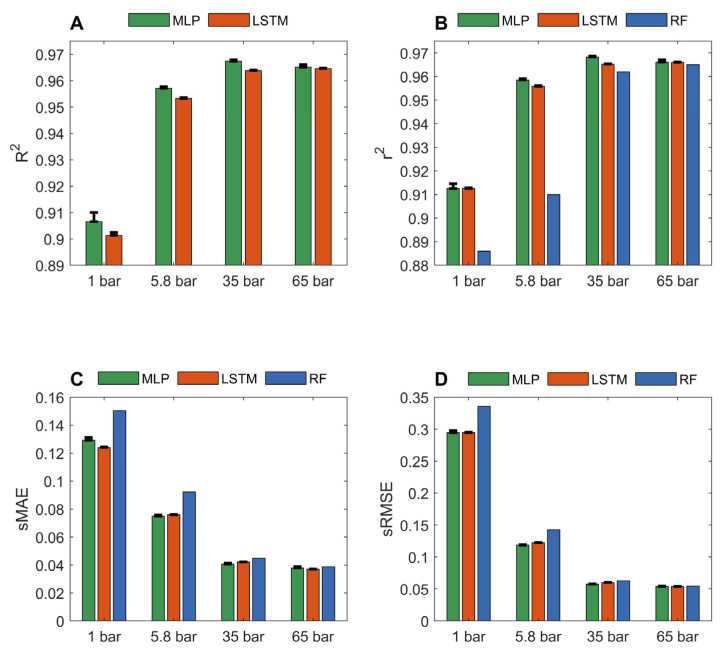
Performance of MLP and LSTM models in the 10-fold cross validations. Pressures at which adsorptions were predicted are marked by x-axis tick labels. Performance was measured in *R*^2^ (**A**); *r*^2^ (**B**); sMAE (**C**); and sRMSE (**D**) and depicted on the y-axes. The average performance measurement values are given in red and green bars for LSTM and MLP models, respectively. The corresponding standard deviations are plotted as sticks above the bars. The results of random forest models from the same datasets reported in Fanourgakis et al.’s JACS paper [36] are plotted as the blue bars.

**Figure 2 nanomaterials-12-03376-f002:**
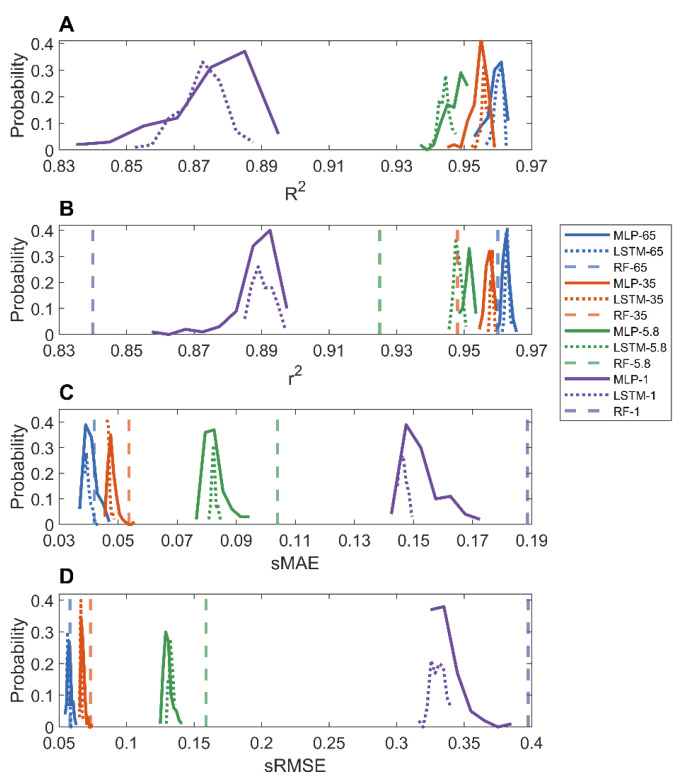
Distributions of *R*^2^ (**A**); *r*^2^ (**B**); sMAE (**C**); and sRMSE (**D**) from holdout validations. Results of MLP, LSTM, and random forest models were plotted in solid, dotted, and dashed lines, respectively. Predictions of methane adsorption pressures were color coded as purple for 1, green for 5.8, red for 35, and blue for 65 bar.

**Figure 3 nanomaterials-12-03376-f003:**
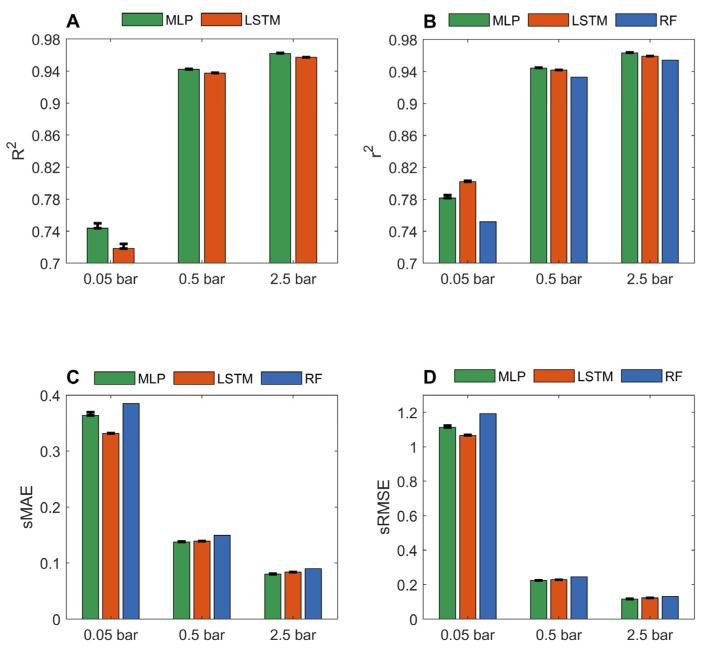
Performance of MLP and LSTM models measured in *R*^2^ (**A**); *r*^2^ (**B**); sMAE (**C**); and sRMSE (**D**). The averaged performance metrics from the 10 iterations of 10-fold cross validations were plotted as green and red bars for MLP and LSTM models, respectively. The corresponding standard deviations were plotted as sticks above the bars. The reported results of random forest models from the same datasets [36] were plotted as blue bars.

**Figure 4 nanomaterials-12-03376-f004:**
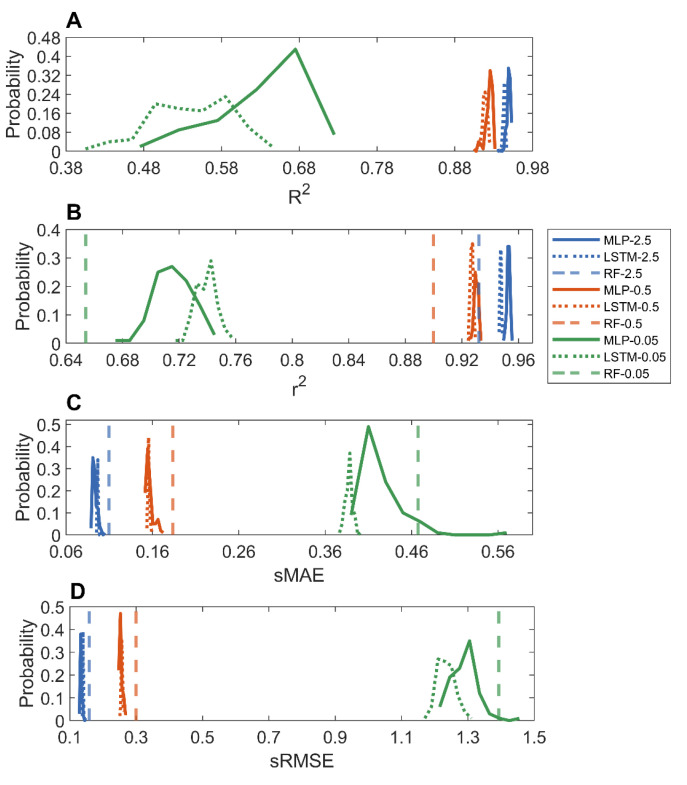
Distributions of performance metrics *R*^2^ (**A**); *r*^2^ (**B**); sMAE (**C**); and sRMSE (**D**) in the 100 holdout validations for MLP (solid curves) and LSTM (dotted curves) models. Pressures of carbon dioxide adsorption predictive models were color coded: green for 0.05 bar, red for 0.5 bar, and blue for 2.5 bar. The x-axis depicts performance metrics marked by the axis label. The reported results of random forest models [36] are shown as vertical lines.

**Figure 5 nanomaterials-12-03376-f005:**
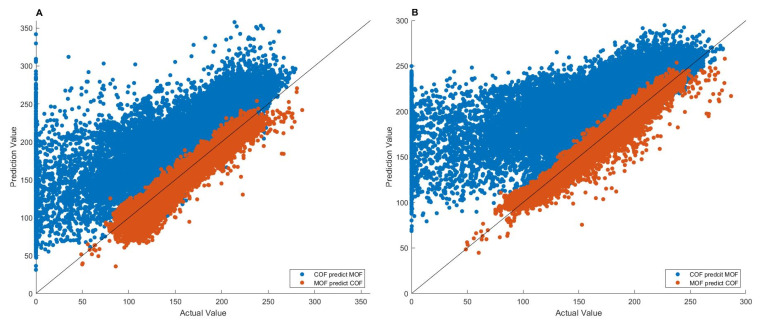
Transferability of MLP (**A**) and LSTM (**B**) models. The x-axis gives actual value and the y-axis depicts predicted value. The values are the volumetric-based adsorption capacities in units of cm^3^(STP)/cm^3^. The blue points are adsorptions of MOFs predicted by models trained on COFs, while the red points are adsorptions of COFs predicted by models trained on MOFs.

**Figure 6 nanomaterials-12-03376-f006:**
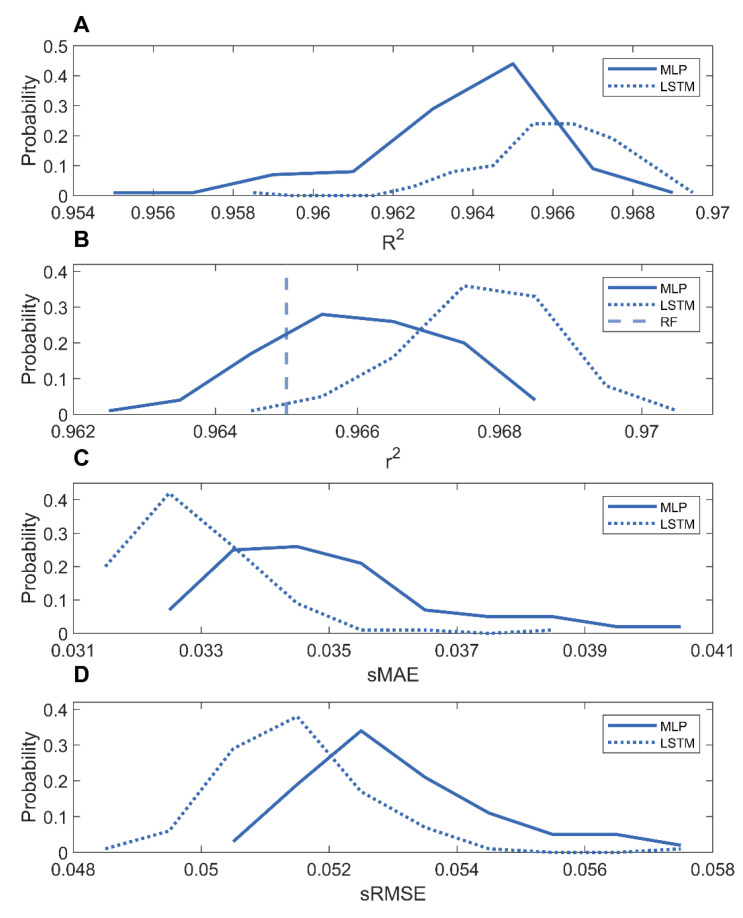
Distributions of performance metrics *r*^2^ (**A**); *R*^2^ (**B**); sMAE (**C**); and sRMSE (**D**) of LSTM (dotted curves) and MLP (solid curves) models trained on the mixture of MOFs and COFs. The reported *r*^2^ of random forest models [36] are shown as the vertical dashed line.

**Table 1 nanomaterials-12-03376-t001:** Datasets used.

Gas	Pressure (bar)	MOFs	Mean Adsorption	Standard Deviation
CO_2_	0.05	70,433	2.2466	5.3255
CO_2_	0.5	70,433	37.365	35.3544
CO_2_	2.5	70,433	92.9512	56.5359
CH_4_	1	70,608	17.8184	17.7097
CH_4_	5.8	28,417	57.5549	33.442
CH_4_	35	70,605	139.1942	44.8701
CH_4_	65	27,151	172.11	50.1996
CH_4_	65	17,098 *	153.3413	37.7033

* Covalent organic frameworks (COFs). The mean adsorption is the volumetric-based adsorption in units of cm^3^(STP)/cm^3^. STP stands for standard temperature and pressure.

## Data Availability

The data supporting the findings of this study are available upon request.

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
