# Peer review of "Deep Learning Models for Predicting Gas Adsorption Capacity of Nanomaterials"

_nanomaterials, 2022, doi:10.3390/nano12193376_

Round 1
Reviewer 1 Report
Discussion is insufficient though results are exhibited. They should discuss the reason why MLP and LSTM models are better than the random forest models at a low pressure or propose the improved mechanism of molecular adsorption on nanomaterials from the results.
Author Response
Discussion is insufficient though results are exhibited. They should discuss the reason why MLP and LSTM models are better than the random forest models at a low pressure or propose the improved mechanism of molecular adsorption on nanomaterials from the results.
Response: We thank the reviewer for this suggestion. Accordingly, we added following sentence to discuss this phenomenon: “At low pressures, the strength of interactions between adsorbent molecules and MOFs plays a more important role in the adsorption capacity of MOFs. We speculate that adsorption capacities of MOFs at high pressures are mainly deter-mined by the shapes of structures of MOFs as well as the accessible pore volumes and internal surface areas of MOFs which are easy to be characterized for MOFs. Therefore, adsorption capacities at high pressures can be accurately predicted using both traditional machine learning and deep learning algorithms. However, the chemical properties such as hydrophobic and electrostatic interactions between gases and MOFs at atomic level contribute more to the adsorption capacities of MOFs for gases at low pressure. The interactions at atomic level between gases and MOFs are more difficult to model. Thus, the methods that can recognize complicate relationships such as deep learning algorithms could better model gas adsorption of MOFs at low pressures.”
Reviewer 2 Report
In the present manuscript the authors used deep learning models to predict the CH4 and CO2 adsorption by MOFs and COFs. Databases of hypothetical materials were employed. The adsorption capacities of these materials at various different thermodynamic conditions have been computed in previous works using molecular simulations.
I have to say that the present manuscript does not bring to our attention any new knowledge from the physical point of view. It is my understanding that from the physical view point all findings are in agreement with a previous work as the authors of present article mention at some points. For example, the usage of atom types as descriptors, the predictions for a type of materials using ML models that were trained on different material types, the lower performance of ML models at low pressures etc., have been already discussed. Some of these findings are appearing as new conclusions (see Conclusion section) in the present manuscript. Also, the need for a different set of descriptors (beyond the atom number density) in order to get improved predictions in the low pressure regime has been nowadays recognized and there are several works that are using energy-based descriptors (see for example works of Fanourgakis, Bucior and Zhiwei Qiao).
On the other hand the predictive performance of the present ML algorithms appears to be higher than that of the Random Forest (RF) algorithm that was used in the original work by Fanourgakis et al. It is encouraging that in all different predictive models created, the performance of Deep Learning models is systematically higher than that of the RF. Personally, after reading the present manuscript I will consider these algorithms (MLP and LSTM) in my feature research. Based on that fact I have to accept that the present manuscript provide a useful information. The manuscript is overall clearly and well written. I have to ask the authors to discuss in more detail (in particular in the Conclusions section) how their physical findings are compared with that of the original work, before I suggest publication of the present manuscript.
Author Response
In the present manuscript the authors used deep learning models to predict the CH4 and CO2 adsorption by MOFs and COFs. Databases of hypothetical materials were employed. The adsorption capacities of these materials at various different thermodynamic conditions have been computed in previous works using molecular simulations.
I have to say that the present manuscript does not bring to our attention any new knowledge from the physical point of view. It is my understanding that from the physical view point all findings are in agreement with a previous work as the authors of present article mention at some points. For example, the usage of atom types as descriptors, the predictions for a type of materials using ML models that were trained on different material types, the lower performance of ML models at low pressures etc., have been already discussed. Some of these findings are appearing as new conclusions (see Conclusion section) in the present manuscript. Also, the need for a different set of descriptors (beyond the atom number density) in order to get improved predictions in the low pressure regime has been nowadays recognized and there are several works that are using energy-based descriptors (see for example works of Fanourgakis, Bucior and Zhiwei Qiao).
Response: We appreciate the reviewer for the criticism. We agree with the reviewer that our study does not bring new knowledge from the physical point of view which is not our study objective. Our work mainly aimed at investigating advantage of deep learning over traditional machine learning for predicting gas adsorption of MOFs. Yes, we confirmed the findings in the previous work by Fanourgakis et al. that the new descriptors significantly improved the prediction of gas adsorption in MOFs using traditional machine learning algorithm, random forest. However, our main finding is that deep learning models (LSTM and MLP) further improve the prediction accuracy.
We agree with the reviewer that recent studies[1-3] have proposed the use of energetic descriptors to improve the predictions in the low-pressure regime. For example, Fanourgakis et al.[2] proposed the use of potential energy surface of a system and evaluate the performance on 4764 computation-ready, experimental (CoRE) MOFs. Bucior et al.[1] used sorbate-sorbent energy histograms as descriptors and tested on more than 50,000 experimental MOFs. Deng et al.[3] used heat of adsorption as the energy descriptors to screen 6,013 CoRE-MOFs. All these studies showed the great potential of using energy descriptors in the MOFs studies. However, these findings are not conflict with our finding that deep learning improves prediction accuracy for adsorption capacities of MOFs. We expect that the deep learning models would be improve when the energetic descriptors are available to the data sets.
We accepted the reviewer’s suggestion and added following sentences to discuss it: “Energy descriptors have been used in recent studies [34,35,53] to improve predictions in the low-pressure regime. For example, Fanourgakis et al. [35] used potential energy surface as descriptors and evaluated the model performance on 4764 computation-ready, experimental (CoRE) MOFs. Bucior et al. [53] used sorbate-sorbent energy histograms as descriptors and tested on more than 50,000 experimental MOFs. Deng et al. [34] used heat of adsorption as the energy descriptors to screen 6,013 CoRE-MOFs. Therefore, the deep learning models for predicting gas adsorption of MOFs at low pressures are expected to be improved by including chemical and energetic descriptors.”
On the other hand the predictive performance of the present ML algorithms appears to be higher than that of the Random Forest (RF) algorithm that was used in the original work by Fanourgakis et al. It is encouraging that in all different predictive models created, the performance of Deep Learning models is systematically higher than that of the RF. Personally, after reading the present manuscript I will consider these algorithms (MLP and LSTM) in my feature research. Based on that fact I have to accept that the present manuscript provide a useful information. The manuscript is overall clearly and well written. I have to ask the authors to discuss in more detail (in particular in the Conclusions section) how their physical findings are compared with that of the original work, before I suggest publication of the present manuscript.
Response: We appreciate the reviewer for the encouraging comment and agree with our main finding that deep learning outperformed traditional machine learning (random forest). We also thank the reviewer for this great suggestion to discuss our findings and that of the original work by Fanourgakis et al. in the Conclusion section. The following sentences added to the Conclusion section of our revised manuscript:
“In the previous work of Fanourgakis et al. [36], they introduced a new chemical descriptor “atom type’ in the machine learning algorithms to account for the chemical characters of MOFs. They evaluated the traditional machine learning models (random forest) on different gases (carbon dioxide, methane) and found using the new descriptors could significantly improve the prediction accuracy. In our work, we took a step further to investigate whether deep learning models could further improve the prediction of gas adsorption capacities. We developed MLP and LSTM models to predict gas adsorption capacities in a large number of topologically diverse MOFs and COFs. Both structural descriptors and chemical descriptors were used to characterize the MOFs and COFs. The performance of the deep learning models built with MLP and LSTM were evaluated using 10-fold cross validations and holdout validations. Our results confirmed that models with both structural and chemical descriptors could accurately predict the gas adsorption capacities of MOFs. We also found that deep learning models can further improve the prediction accuracy by handling the complex relationship between gas adsorption capacities and MOFs structures, especially at low pressures. “
Reference
- Bucior, B.J.; Bobbitt, N.S.; Islamoglu, T.; Goswami, S.; Gopalan, A.; Yildirim, T.; Farha, O.K.; Bagheri, N.; Snurr, R.Q. Energy-based descriptors to rapidly predict hydrogen storage in metal–organic frameworks. Molecular Systems Design & Engineering 2019, 4, 162-174, doi:10.1039/C8ME00050F.
- Fanourgakis, G.S.; Gkagkas, K.; Tylianakis, E.; Klontzas, E.; Froudakis, G. A Robust Machine Learning Algorithm for the Prediction of Methane Adsorption in Nanoporous Materials. The Journal of Physical Chemistry A 2019, 123, 6080-6087, doi:10.1021/acs.jpca.9b03290.
- Deng, X.; Yang, W.; Li, S.; Liang, H.; Shi, Z.; Qiao, Z. Large-Scale Screening and Machine Learning to Predict the Computation-Ready, Experimental Metal-Organic Frameworks for CO2 Capture from Air. Applied Sciences 2020, 10, 569.
- Fanourgakis, G.S.; Gkagkas, K.; Tylianakis, E.; Froudakis, G.E. A Universal Machine Learning Algorithm for Large-Scale Screening of Materials. J. Am. Chem. Soc. 2020, 142, 3814-3822, doi:10.1021/jacs.9b11084.
Reviewer 3 Report
The authors present a ML study of adsorption of CH4 and CO2 in MOFs and COFs based on hypothetical materials databases with simulated (GCMC) adsorption data. They identify that deep learning methods MLP and LSTM outperform conventional ML methods for predictive data, and identify that high-pressure data are better predicted on the basis of the structural descriptors to which they fit. They identify that a better set of structure/chemistry descriptors may help improve their models in areas where they are lacking.
The demonstration in this work that their employed ML approaches outperform other ML approaches represents what I see as the most significant achievement in this work, and this is sufficient to warrant publication of the work, although it could be substantially strengthened. Numerous other features of the work and comparisons to make this study more broadly useful to the MOF and COF research communities, however, are lacking and could stand to be improved in revision of the manuscript.
There are also numerous technical errors and omissions in this work that must be addressed before this manuscript is suitable for publication – most of these are quite minor and range from typographical errors to technical errors to simply incomplete representations of information that prevent meaningful appreciation of data. I’ve enumerated these below:
Typographical: carbon dioxide is at least once referred to as “carbon”, adsorption is at least once referred to as “absorption”, formatting of reference citations in the manuscript is inconsistent (superscript/bracket) and should be made uniform.
Technical: carbon dioxide does not have a dipole moment, and electrostatic interactions are driven by quadrupolar interaction. This is incorrectly reported as dipolar interaction (or something else entirely is meant when carbon dioxide’s dipole is discussed in the paper). Nonetheless, this must be corrected or better explained before publication.
Omissions: Most figures report an adsorption amount but do not report units for adsorption. Typical units may be mmol/mol, mmol/g, or similar, but it is unclear what is used and that makes interpretation of the magnitudes of the deviations represented in figures s4-s33, table 1, difficult.
More generally, the authors highlight the higher computational cost of GCMC in characterizing materials as motivating ML approaches. GCMC is force-field based, and for MOFs generally there has historically been a major challenge of developing accurate force-fields for describing adsorption. Especially challenging in this regard is the Henry regime, which is the exact (low pressure) regime that this work identifies as challenged by the authors’ method, a method that consists of fitting referencing the authors’ ML work to GCMC using general force fields. This is especially a challenge for some of the very chemical features that make many MOFs of interest for adsorption/separation applications, namely the undercoordinated metals that many MOF structures feature. Therefore, it seems extremely likely that many attractive MOFs would be missed in screening using the authors’ methods. The authors mention as much in their conclusions about training on COFs vs MOF sand transferability of their fits, but don’t address the underlying challenge of a poorly modeled Henry regime. The authors' conclusion that their methods are likely useful for high pressure adsorption are well founded on the basis of the data shown. Given that the comparisons are based on force fields, the reference data used as “actual” is very likely to have significant error in the low pressure regime already, potentially making it a poor choice of reference. The data fit well in the saturation limit, but this already is fairly well modeled by conventional force fields, so this work only represents a savings in cost in screening materials in this regard. If the goal is to demonstrate cost savings in modeling the high pressure regime, however, it is unclear that this will outperform a more rudimentary method based simply on internal surface areas and accessible pore volumes, such as the information available through, i.e., Zeo++. If the goal is to demonstrate cost savings over GCMC or other commonly employed methods, it may be useful to benchmark these methods or compute performance/cost to compare.
In figures s4 – s33 there are some wild disagreements between ML-predicted and GCMC-predicated data without any attribution to what may be giving rise to these disagreements. Especially instances where the model vastly overpredicts adsorption of methane at modest pressures (where uptake is mostly driven by dispersion) this is a concern for the reliability of this method. It nonetheless may be helpful in screening a large dataset, as the authors report. It would be interesting to see what in the authors’ models are leading to these failures... the GCMC reference data should do an adequate job of modeling dispersion due to a Lennard-Jones-like term in computing energetics using force-fields.
Overall, the work represents a useful contribution in choice of ML method for analyzing adsorbent datasets, but the reliability of the dataset used (GCMC/FF on hypothetical MOFs/COFs) presents concerns for reliability of the predictions based on fitting to those data. A better training set should inevitably lead to availability of better fits. The leading argument of cost savings in high-throughput screening of large databases is salient, but the paper would benefit from cost benchmarking and a broader comparison of methods… the current comparison of authors’ ML method vs conventional ML and GCMC is very limited. It would be interesting to see how the predictions regress against structural features relevant for saturation loading such as adsorption surface area or accessible volume to better understand how this approach may augment the low-cost screening approaches available for candidate storage materials.
Author Response
The authors present a ML study of adsorption of CH4 and CO2 in MOFs and COFs based on hypothetical materials databases with simulated (GCMC) adsorption data. They identify that deep learning methods MLP and LSTM outperform conventional ML methods for predictive data, and identify that high-pressure data are better predicted on the basis of the structural descriptors to which they fit. They identify that a better set of structure/chemistry descriptors may help improve their models in areas where they are lacking.
The demonstration in this work that their employed ML approaches outperform other ML approaches represents what I see as the most significant achievement in this work, and this is sufficient to warrant publication of the work, although it could be substantially strengthened. Numerous other features of the work and comparisons to make this study more broadly useful to the MOF and COF research communities, however, are lacking and could stand to be improved in revision of the manuscript.
There are also numerous technical errors and omissions in this work that must be addressed before this manuscript is suitable for publication – most of these are quite minor and range from typographical errors to technical errors to simply incomplete representations of information that prevent meaningful appreciation of data. I’ve enumerated these below:
Typographical: carbon dioxide is at least once referred to as “carbon”, adsorption is at least once referred to as “absorption”, formatting of reference citations in the manuscript is inconsistent (superscript/bracket) and should be made uniform.
Response: We appreciate the reviewer for the careful review and for catching the typos. We corrected these typos in revision and checked the revised manuscript carefully.
Technical: carbon dioxide does not have a dipole moment, and electrostatic interactions are driven by quadrupolar interaction. This is incorrectly reported as dipolar interaction (or something else entirely is meant when carbon dioxide’s dipole is discussed in the paper). Nonetheless, this must be corrected or better explained before publication.
Response: We thank the reviewer for catching this error. The incorrect term “dipole moment” was replaced with “quadrupolar interactions” in our revised manuscript.
Omissions: Most figures report an adsorption amount but do not report units for adsorption. Typical units may be mmol/mol, mmol/g, or similar, but it is unclear what is used and that makes interpretation of the magnitudes of the deviations represented in figures s4-s33, table 1, difficult.
Response: We are sorry for missing the units for adsorption capacity in our original submission. The adsorption capacities in the datasets are volumetric based, and the units are cm3(STP) / cm3. We added units in the figures and tables.
More generally, the authors highlight the higher computational cost of GCMC in characterizing materials as motivating ML approaches. GCMC is force-field based, and for MOFs generally there has historically been a major challenge of developing accurate force-fields for describing adsorption. Especially challenging in this regard is the Henry regime, which is the exact (low pressure) regime that this work identifies as challenged by the authors’ method, a method that consists of fitting referencing the authors’ ML work to GCMC using general force fields. This is especially a challenge for some of the very chemical features that make many MOFs of interest for adsorption/separation applications, namely the undercoordinated metals that many MOF structures feature. Therefore, it seems extremely likely that many attractive MOFs would be missed in screening using the authors’ methods. The authors mention as much in their conclusions about training on COFs vs MOF s and transferability of their fits, but don’t address the underlying challenge of a poorly modeled Henry regime. The authors' conclusion that their methods are likely useful for high pressure adsorption are well founded on the basis of the data shown. Given that the comparisons are based on force fields, the reference data used as “actual” is very likely to have significant error in the low pressure regime already, potentially making it a poor choice of reference. The data fit well in the saturation limit, but this already is fairly well modeled by conventional force fields, so this work only represents a savings in cost in screening materials in this regard. If the goal is to demonstrate cost savings in modeling the high pressure regime, however, it is unclear that this will outperform a more rudimentary method based simply on internal surface areas and accessible pore volumes, such as the information available through, i.e., Zeo++. If the goal is to demonstrate cost savings over GCMC or other commonly employed methods, it may be useful to benchmark these methods or compute performance/cost to compare.
Response: We appreciate the reviewer for the constructive comments. We are sorry for not having stated our goal clearly and, thus, misleading the reviewer. Cost saving for machine learning and deep learning over GCMC is well recognized in the field. The goal of our study is to demonstrate that deep learning outperforms traditional machine learning in the prediction of gas adsorption capacities of MOFs. We agree with the reviewer on the limitations of the findings using the adsorption capacities calculated with GCMC which is a force-field based method on the Henry’s law, especially at low pressures. Accordingly, we added following paragraph in our revision to discuss it.
“Another challenge in predicting adsorption capacities of MOFs using machine learning and dee learning is the lack of experimental data for training. Experimentally determined adsorption capacities are the best for training machine learning and deep learning models. However, to experimentally generate adsorption data for such a large number of MOFs is time consuming and costly, making it very practically challenging if not impossible. Therefore, most of the current practices use the adsorption capacities calculated by GCMC which is a force-field method based on the Henry’s law. There has historically been a major challenge of developing accurate force-fields for describing adsorption, especially for adsorption at low pressures based on the Henry’s law. In brief, Henry’s law states that the adsorption capacity of an MOF is proportional to the gas pressure at a constant which can be determined experimentally or with a force-filed based method. Usually, at high pressures, the effect of gas pressure overpasses the contributions from chemical and energetic interactions between the MOF and the gas. Therefore, the adsorption capacities at high pressures estimated from a force-field based method are close to that determined by experiments. However, at low pressures, chemical and energetic interactions between the MOF and the gas are important to the adsorption capacity and are not linear proportional to the gas pressure. Thus, the adsorption capacities at low pressures estimated using a general force-field method may have a large difference from that determined experimentally. Though the deep learning models much improved the prediction performance on adsorption capacities at low pressures, it should be cautious in the utilization of the results because the predicted values are fitted to the adsorption capacities estimated by GCMC but not to that determined experimentally.”
We like the reviewer’s suggestion to check if the deep learning models could be improved when including more rudimentary information such as internal surface areas and accessible pore volumes. The surface areas were contained in our datasets. Therefore, according to the reviewer’s suggestion, we calculated the accessible pore volumes using Zeo++ with the probe radius of 1.625 Å. We then developed and validated the MLP and LSTM models for predicting methane adsorption at 1 bar by integrating the accessible pore volumes into the datasets and using the same learning protocol. The results were summarized in supplementary Figures S20 and S21 and compared with our original models and models without atom types in the supplementary Table S3 which were added in our revision and are given below. As it can be seen from the comparison (Table S3), the descriptors of atom types are important for predicting methane adsorptions at low pressures because the models without atom types much underperformed, while the inclusion of accessible volumes did not improve the deep learning models. Because accessible volumes are correlated with surface areas, our results indicate that adding correlated descriptors would not improve deep learning models for predicting gas adsorption of MOFs. Following paragraph was added in our revision to discuss it.
“To examine if the deep learning models could be improved when including more rudimentary information such as accessible pore volumes. We calculated the accessible pore volumes using Zeo++ (http://zeoplusplus.org/) with the probe radius of 1.625 Å. We then developed MLP and LSTM models for predicting methane adsorption at 1 bar by integrating the accessible pore volumes into the datasets and using the same learning protocol. We also developed MLP and LSTM models using datasets with removal of atom types and inclusion of accessible pore volumes. The results from 100 holdout validations on the models with all 26 descriptors were summarized in supplementary Figures S20 and S21 and the models developed with different sets of descriptors were compared in the supplementary Table S3. As it can be seen from the comparison, the descriptors of atom types are important for predicting methane adsorptions at low pressures because the models without atom types much underperformed, while the inclusion of accessible volumes did not improve the deep learning models. Because accessible volumes are correlated with surface areas, our results indicate that adding correlated descriptors would not improve deep learning models for predicting gas adsorption of MOFs.”
Table S3: Holdout validations performance of models with and without accessible pore volumes for predicting methane adsorption at 1 bar
model |
Number of descriptors* |
R2 |
r2 |
sMAE |
sRMSE |
LSTM |
26 |
0.8649±0.0064 |
0.8866±0.0031 |
0.1492±0.0014 |
0.3361±0.0055 |
LSTM |
25 |
0.8722±0.0066 |
0.8902±0.0030 |
0.1464±0.0013 |
0.3306±0.0052 |
LSTM |
6 |
0.6148±0.0239 |
0.7406±0.0034 |
0.2894±0.0018 |
0.5109±0.0053 |
MLP |
26 |
0.8800±0.0071 |
0.8919±0.0030 |
0.1550±0.0056 |
0.3293±0.0055 |
MLP |
25 |
0.8755±0.0120 |
0.8891±0.0063 |
0.1526±0.0063 |
0.3350±0.0103 |
MLP |
6 |
0.6951±0.0335 |
0.7748±0.0056 |
0.2760±0.0112 |
0.4773±0.0089 |
*5 descriptors: the dominant pore size, the max pore size, the void fraction, the gravimetric surface area, density; 6 descriptors: the 5 descriptors plus accessible volume; 25 descriptors: the 5 descriptors plus the 20 atom types; 26 descriptors: the 6 descriptors plus the 20 atom types.
In figures s4 – s33 there are some wild disagreements between ML-predicted and GCMC-predicated data without any attribution to what may be giving rise to these disagreements. Especially instances where the model vastly overpredicts adsorption of methane at modest pressures (where uptake is mostly driven by dispersion) this is a concern for the reliability of this method. It nonetheless may be helpful in screening a large dataset, as the authors report. It would be interesting to see what in the authors’ models are leading to these failures... the GCMC reference data should do an adequate job of modeling dispersion due to a Lennard-Jones-like term in computing energetics using force-fields.
Response: We appreciate the reviewer for raising this great point with sharp eyes. According to the reviewer’s suggestion, we had a close look up at the MOFs having large differences between the deep learning models and GCMC-predicted adsorptions at modest pressures. We found that MOFs with smaller accessible pores, less surface area, and higher weight density had larger disagreements between the deep learning model predicted and GCMC calculated adsorption capacities. For example, comparing the deep learning prediction methane adsorption capacities at 5.8 bar with the GCMC calculated values (supplementary Figure S16) revealed that the 1,775 MOFs with over 100% overpredictions have average dominant pore diameter 4.35 Å, gravimetric surface area 969.57 m²/g, and weight density 1.37 g/cm3, while the 4,542 MOFs with less than 5% disagreements have average dominant pore diameter 8.76 Å, gravimetric surface area 3267.07 m²/g, and weight density 0.74 g/cm3. As gas adsorption is mostly driven by dispersion at modest pressures, MOFs with great density usually have small pores and surface areas which make gas dispersion more difficult and adsorption capacities are determined by multiple chemical and structural features in a complicated relationship. Therefore, predicting adsorption capacities of such kind MOFs at modest pressures is more challenging than MOFs with low density, large pores, and large surface areas. Though we demonstrated that the developed deep learning models are helpful in screening a large dataset, utilization of prediction results for MOFs with great density, small pores and surface areas should be cautious. Following paragraph was added in our revision to discuss it.
“When examining the MOFs that showed large differences between the deep learning models and GCMC-predicted adsorptions at modest pressures, we found that MOFs with smaller accessible pores, less surface area, and greater weight density had larger disagreements between the deep learning model predicted and GCMC calculated adsorption capacities. For example, comparing the deep learning predicted methane adsorption capacities at 5.8 bar with the GCMC calculated values (supplementary Figure S16) revealed that the 1,775 MOFs with over 100% overpredictions have aver-age dominant pore diameter 4.35 Å, gravimetric surface area 969.57 m²/g, and weight density 1.37 g/cm3, while the 4,542 MOFs with less than 5% disagreements have aver-age dominant pore diameter 8.76 Å, gravimetric surface area 3267.07 m²/g, and weight density 0.74 g/cm3. As gas adsorption is mostly driven by dispersion at modest pressures, MOFs with great density usually have small pores and surface areas which make gas dispersion more difficult and adsorption capacities are determined by multiple chemical and structural features in a complicated relationship. Therefore, predicting adsorption capacities of such kind MOFs at modest pressures is more challenging than MOFs with low density, large pores, and large surface areas. Though we demonstrated that the developed deep learning models are helpful in screening a large dataset, utilization of the deep learning prediction results for MOFs with great density, small pores and surface areas should be cautious.”
Overall, the work represents a useful contribution in choice of ML method for analyzing adsorbent datasets, but the reliability of the dataset used (GCMC/FF on hypothetical MOFs/COFs) presents concerns for reliability of the predictions based on fitting to those data. A better training set should inevitably lead to availability of better fits. The leading argument of cost savings in high-throughput screening of large databases is salient, but the paper would benefit from cost benchmarking and a broader comparison of methods… the current comparison of authors’ ML method vs conventional ML and GCMC is very limited. It would be interesting to see how the predictions regress against structural features relevant for saturation loading such as adsorption surface area or accessible volume to better understand how this approach may augment the low-cost screening approaches available for candidate storage materials.
Response: We appreciate the reviewer for the encouraging comment on the usefulness of our findings that deep learning is suggested for analyzing adsorbent datasets. We realize and agree with the reviewer on the concern on the reliability of the datasets used. We cannot agree more with the reviewer that a better training set should inevitably lead to availability of better fits. As the number of possible MOFs is very large, experimentally determining adsorption capacities is very challenging at current practices. Exploring more accurate calculation methods including machine learning and deep learning are expected to remain as an active research topic in the future. Once again, Calculation cost saving is not our objective and we aimed at comparing traditional machine learning and deep learning using these datasets as case studies. The reviewer’s question on how the predictions regress against structural features relevant for saturation loading such as adsorption surface area or accessible volume is valuable. Following your suggestions, we used Zeo++ to calculate accessible volume. We investigated how the predictions regress against descriptors including the calculated accessible volume. The inclusion of accessible volumes did not improve the deep learning models.
Round 2
Reviewer 1 Report
Dear authors,
The differences from previous works have been cleared. It is shown that MLP and LSTM deep learning models are useful for predicting adsorption capacity of MOFs. However, it is necessary to revise the following.
1. The authors should describe that the method in Ref. 36 showed the best performance in predictions by the conventional machine learning models using the same data set more clearly to say that MLP and LSTM deep learning models are more useful than them. At least, they should describe the rating of the results in Ref.36 by them.
2. The authors describe “… adding correlated descriptors would not improve deep learning models for predicting gas adsorption of MOFs.” in L393-394. On the other hand, they say “… descriptors more related to gas adsorption are needed to improve deep learning models for predicting gas adsorption at low pressure in Conclusions (L578-580). These sentences appear inconsistent.
3. The revised version of your supplemental file cannot be found although tab. S3 and figs. S20 and S21 should be added in it.
4. “Figure S28-S33” in L599 appears to be mistaken for “Figure S28-S35” if figs S20 and 21 are newly inserted in the supplemental file. Please check supplemental figure numbers.
Author Response
Reviewer #1
The differences from previous works have been cleared. It is shown that MLP and LSTM deep learning models are useful for predicting adsorption capacity of MOFs. However, it is necessary to revise the following.
- The authors should describe that the method in Ref. 36 showed the best performance in predictions by the conventional machine learning models using the same data set more clearly to say that MLP and LSTM deep learning models are more useful than them. At least, they should describe the rating of the results in Ref.36 by them.
Response: We thank the reviewer for this suggestion. Accordingly, we added the following sentences to discuss the Ref. 36(Fanourgakis et al.).
“The random forest models developed by Fanourgakis et al. [36] have greatly improved the prediction accuracy of gas adsorptions of MOFs. Their models significantly outperformed previously reported models [35,42] where only structural descriptors were used. By directly comparing our results with the results from the random forest models [36], we demonstrated the potential of our deep learning models to accurately predict the gas adsorption capacities of MOFs.“
- The authors describe “… adding correlated descriptors would not improve deep learning models for predicting gas adsorption of MOFs.” in L393-394. On the other hand, they say “… descriptors more related to gas adsorption are needed to improve deep learning models for predicting gas adsorption at low pressure in Conclusions (L578-580). These sentences appear inconsistent.
Response: We thank the reviewer for the careful review of our manuscript. The descriptors discussed at these two places are not the same. In L393-394, “correlated descriptors” are referred to descriptors that are correlated each other. For example, accessible volume and surface areas are correlated descriptors because they both can be used to describe the volume of pores. Therefore, adding accessible volume as the new descriptor would not provide much extra information regarding MOFs structures and the deep learning model performance would not be much improved. In Line578-580, “the descriptors more related to gas adsorption” are referred to the independent and uncorrelated descriptors that more related to the gas adsorption. Since these descriptors can provide the new information regarding gas adsorption, they can be used to improve our deep learning models for predicting gas adsorption.
We added the word “much” in L393-394 and “Independent’ in L578-580 to be more accurate.
“Because accessible volumes are correlated with surface areas, our results indicate that adding correlated descriptors would not much improve deep learning models for predicting gas adsorption of MOFs.”
“Therefore, independent descriptors more related to gas adsorption are needed to improve deep learning models for predicting gas adsorption at low pressure.”
- The revised version of your supplemental file cannot be found although tab. S3 and figs. S20 and S21 should be added in it.
Response: We thank the reviewer for noticing this. We uploaded the revised supplementary file, but the system still shows the previous old version. We are unclear about what the cause is. We uploaded the revised supplementary file again and double checked to make sure the new version is showed in the system.
- “Figure S28-S33” in L599 appears to be mistaken for “Figure S28-S35” if figs S20 and 21 are newly inserted in the supplemental file. Please check supplemental figure numbers.
Response: We appreciate the reviewer for the careful review. The supplementary file you looked at is the old one. Sorry for this. We checked the revised supplementary figures and “Figure S28-S33” are correct. After adding the two new figures S20 and 21, the old Figure S26-S31 now become Figure S28-S33.

Reviewer 2 Report
I believe that the revised manuscript addresses all concerns I had, and it is significantly improved after considering the excellent suggestions of the other reviewer as well. I suggest its publication.
PS. line 298: correct "dee"->"deep"
Author Response
I believe that the revised manuscript addresses all concerns I had, and it is significantly improved after considering the excellent suggestions of the other reviewer as well. I suggest its publication.
Response: Thank you for agreeing to accept our manuscript. Your constructive comments and suggestions have greatly improved our manuscript.
- line 298: correct "dee"->"deep"
Response: We appreciate the reviewer for the careful review and for catching the typo. We corrected this typo in revision and checked the revised manuscript carefully.
